# Design, Preparation, and Evaluation of a Novel Elastomer with Bio-Based Diethyl Itaconate Aiming at High-Temperature Oil Resistance

**DOI:** 10.3390/polym11111897

**Published:** 2019-11-17

**Authors:** Hui Yang, Haijun Ji, Xinxin Zhou, Weiwei Lei, Liqun Zhang, Runguo Wang

**Affiliations:** 1Beijing Advanced Innovation Center for Soft Matter Science and Engineering, State Key Laboratory of Organic-Inorganic Composites, Beijing Laboratory of Biomedical Materials, Beijing University of Chemical Technology, Beijing 100029, China; 2017310124@mail.buct.edu.cn (H.Y.); 2017210203@mail.buct.edu.cn (H.J.); zhanglq@mail.buct.edu.cn (L.Z.); 2Hubei Key Laboratory of Polymer Materials, School of Materials Science & Engineering, Hubei University, Wuhan 430062, China

**Keywords:** bio-based diethyl itaconate, redox emulsion polymerization, composite, mechanical properties, high-temperature oil-resistant

## Abstract

A novel elastomer poly(diethyl itaconate-*co*-butyl acrylate-*co*-ethyl acrylate-*co*-glycidyl methacrylate) (PDEBEG) was designed and synthesized by redox emulsion polymerization based on bio-based diethyl itaconate, butyl acrylate, ethyl acrylate, and glycidyl methacrylate. The PDEBEG has a number average molecular weight of more than 200,000 and the yield is up to 96%. It is easy to control the glass transition temperature of the PDEBEG, which is ranged from −25.2 to −0.8 °C, by adjusting the monomer ratio. We prepared PDEBEG/CB composites by mixing PDEBEG with carbon black N330 and studied the oil resistance of the composites. The results show that the tensile strength and the elongation at break of the composites with 10 wt% diethyl itaconate can reach up to 14.5 MPa and 305%, respectively. The mechanical properties and high-temperature oil resistance of the composites are superior to that of the commercially available acrylate rubber AR72LS.

## 1. Introduction

Rubber material is an important strategic material for the national economy and the national defense industry, among which oil-resistant rubber takes a vital role. It is widely used to manufacture oil-resistant products in the automobile manufacturing, aerospace, and machinery industries, such as oil pipes, drums, and gaskets [1]. At present, the most commonly used rubber in oil seal materials is nitrile–butadiene rubber (NBR) [2], which has a good oil resistance, wear resistance, and weather resistance [3,4]. With the development of industrialization, especially in the automotive industry, the requirements for rubber oil resistance and heat resistance are getting more and more strict. In order to meet these requirements, rubber material has also undergone a great change from NBR to acrylate rubber (ACM) [5], methyl vinyl silicone rubber (MVQ) [6], fluororubber (FKM) [7], hydrogenated nitrile–butadiene rubber (HNBR) [8,9], and so on. 

ACM is an elastomer obtained by copolymerization of acrylate monomers. It has excellent heat resistance, aging resistance, weather resistance, and UV resistance due to a saturated main chain structure [10]. Polar ester groups of side chains make ACM rubber oil-resistant. ACM is superior to NBR in heat resistance and aging resistance [11,12,13]. Compared with MVQ and FKM, ACM has better processing and mechanical properties. Furthermore, ACM has a relative low price, which is only 1/5, 1/3, and 1/12 of the price of HNBR, MVQ, and FKM, respectively. Therefore, in recent years, ACM has been widely used and developed as an oil seal rubber [14].

The monomers used to synthesize oil-resistant rubbers are mainly derived from fossil resources. In order to reduce the dependence on petrochemical resources and maintain sustainable development in the rubber industry [15,16], we urgently need to develop elastomers with bio-based content and engineering application potential to replace full petroleum-based elastomers [17,18,19,20,21]. At present, biochemicals capable of mass production mainly include: 1,3-propanediol, 1,4-butanediol [22,23], itaconic acid [24,25], sebacic acid [26], etc. Itaconic acid is a mass-product by fermentation from carbohydrates using *Aspergillus terreus* [27,28]. It has two carboxyl groups and one double bond in one molecule. As an unsaturated dicarboxylic organic acid, itaconic acid is capable of transforming into various derivatives. Itaconic acid and its derivatives, such as dialkyl itaconate, are expected to replace acrylic or methacrylic monomer to synthesize polymers. The functional carboxylic groups allow itaconic acid to undergo polycondensation with diols and diamines to prepare polyesters [29,30,31,32,33] and polyamides [34,35], respectively. By using the double bonds, itaconic acid can undergo free radical polymerization to prepare poly(itaconic acid). More commonly, dialkyl itaconate is used as a substitute for itaconic acid to homopolymerize or copolymerize with other unsaturated monomers by emulsion polymerization [24,36,37,38]. This emulsion polymerization, in which an isolated latex particle is formed by a micelle mechanism in an aqueous medium-based emulsion and a free polymerization reaction is carried out therein to produce a high polymer, is a green polymerization method. Nowadays, it includes surface-functionalized latexes [39], using a surfactant and organic solvent-free emulsion system [40], using lignin-based polymeric surfactants [41] and Disponil AFX non-ionic surfactants [42], which has been used widely by green emulsion polymerization. Lei et al. [43] synthesized bio-based elastomer PDEII with diethyl itaconate and isoprene and studied the oil resistance of the elastomer with various monomer ratios at different temperatures. The results showed that the high-temperature oil resistance and the thermal-oxidative aging performances of the elastomer deteriorated with the increase of the isoprene content, while the mechanical strength of the elastomer decreased with the decrease of the isoprene content. In addition, the introduction of isoprene provides the crosslink points and softens the macromolecule chains. In order to balance the high-temperature oil resistance, thermal-oxidative aging, and good elasticity of the elastomer with the crosslink points, the monomer isoprene should be replaced by other chemicals. In previous studies, the homopolymer of ethyl acrylate exhibited an excellent high-temperature oil resistance, while the homopolymer of butyl acrylate exhibited a good elasticity [1]. Furthermore, the monomers containing epoxy or active chlorine groups such as glycidyl methacrylate can be used to provide cross-linking points. Thus, a mixture of the ethyl acrylate, butyl acrylate, and glycidyl methacrylate is expected to replace the isoprene to copolymerize with diethyl itaconate, resulting in a novel elastomer with balanced overall properties. 

In this work, a novel elastomer poly(diethyl itaconate-*co*-butyl acrylate-*co*-ethyl acrylate-*co*-glycidyl methacrylate) (PDEBEG) with bio-based content was synthesized by a redox emulsion polymerization. The butyl acrylate endows the PDEBEG with a low-temperature resistance. The mass ratio of diethyl itaconate to ethyl acrylate can adjust the mechanical properties and oil resistance performance of the PDEBEG. The oil resistance of the PDEBEG with various monomer ratios was investigated. In addition, a commercially available acrylate rubber AR72LS was used as a reference to evaluate the mechanical properties and oil resistance of the PDEBEG. The PDEBEG elastomer is expected to be used for seals, radiators, and hoses. The design and preparation route of the novel oil-resistant elastomer is shown in Figure 1.

## 2. Materials and Methods

### 2.1. Materials

Diethyl itaconate (DEI, purity of 98%) was purchased from Sigma-Aldrich Co. (Darmstadt, Germany). Ethyl acrylate (EA, purity of 99%) was purchased from Aladdin Industrial Co. (Shanghai, China). Glycidyl methacrylate (GMA, purity of 97%) was purchased from Aladdin Industrial Co. (Shanghai, China) and passed through a neutral alumina column before use. Butyl acrylate (BA, purity of 99%) was purchased from Aladdin Industrial Co. (Shanghai, China) and purified by distillation under reduced pressure. Sodium dodecyl benzenesulfonate (SDBS, purity of 95%) was purchased from Aladdin Industrial Co. (Shanghai, China) Ferric ethylene diamine tetraacetic acid salt (Fe-EDTA), sodium hydroxymethanesulfinate (SHS), tert-butyl hydroperoxide (TBH), and hydroxylamine (HA) were bought from Sigma-Aldrich Co. (Darmstadt, Germany) and used as received. Acrylate rubber AR72LS was purchased from ZEON Co. (Tokyo, Japan). Standard Oil 3# for ASTM oil resistance performance test of elastomer was supplied by Japan Sun Oil (Tokyo, Japan).

### 2.2. Synthesis of Poly(diethyl itaconate/butyl acrylate/ethyl acrylate/glycidyl methacrylate) (PDEBEG)

The synthesis of the PDEBEG was carried out using the mass ratios of DEI/EA/BA/GMA shown in Table 1. The polymerization reaction is shown in Figure 2. Deionized water (250.0 g, provided by Beijing University of Chemical Technology, Beijing, China), SDBS (3.2 g), Fe-EDTA solution (8.0 g, concentration of aqueous solution was 1.0 wt%), SHS solution (4.0 g, concentration of aqueous solution was 10.0 wt%), and a mixture of the comonomers (DEI, EA, BA, and GMA) were added into a three-neck glass flask under nitrogen atmosphere, and then the emulsion system was stirred at 400 rpm for 1 h. Once the initiator TBH was injected into the flask, the stirring rate was reduced to 210 rpm. After 8 h, the polymerization reaction was terminated by HA (0.4 g), and the PDEBEG latex was obtained. The PDEBEG latex was then coagulated by ethanol to obtain the wet PDEBEG elastomer. It was washed by ethanol to remove impurities and then dried in a vacuum oven at 60 °C for 24 h to obtain the PDEBEG.

### 2.3. Preparation of the PDEBEG/CB Composites

The compounding formula of the PDEBEG/CB composites is shown in Table 2. First, carbon black (CB) was kneaded with the PDEBEG in a Haake internal mixer for 8 min to obtain a mixture. Second, the other additives were kneaded with the mixture to obtain PDEBEG/CB compound. Finally, the PDEBEG/CB compound was cured under 15 MPa at 180 °C to prepare PDEBEG/CB composite. The cure time is determined by a rotor-less vulcanizer.

### 2.4. Measurements and Characterization

#### 2.4.1. Gel Permeation Chromatography (GPC)

The molecular weights of PDEBEG were determined by GPC measurements on a Waters Breeze instrument equipped with three water columns (Styragel HT3_HT5_HT6E) using tetrahydrofuran (THF) as the eluent (1.0 mL/min) and a Waters 2410 refractive index detector (Water, Milford, MA, USA). A polystyrene standard was used for calibration.

#### 2.4.2. Fourier Transform Infrared Spectroscopy (FT-IR)

Fourier transform infrared spectroscopy (FT-IR) analysis was performed on an FT-IT spectrometer (Tensor 27, Bruker, Rheinstetten, Germany). The scan range was 4000–600 cm^−1^ with resolution of 4 cm^−1^. 

#### 2.4.3. Proton Nuclear Magnetic Resonance (^1^H NMR) Spectra

Proton nuclear magnetic resonance (^1^H NMR) spectra of PDEBEG with deuterated chloroform (CDCl_3_) as the solvent were recorded on a Bruker AV400 (Rheinstetten, Germany) spectrometer at room temperature. 

#### 2.4.4. Thermal Performance

Different scanning calorimetry (DSC) measurements were performed with a Mettler Toledo (Greifensee, Switzerland) DSC instrument under a nitrogen atmosphere at a heating rate of 10 °C/min from −80 to 150 °C. Thermal stability was evaluated by thermogravimetric analysis (TGA, Mettler Toledo International Inc., Greifensee, Switzerland). Samples of 10 mg were heated from 30 to 600 °C at a heating rate of 10 °C/min under nitrogen atmosphere.

#### 2.4.5. Curing Characteristics

The curing characteristics of the PDEBEG/CB compounds were measured at 180 °C using a P3555B2 disc vulkameter (Beijing Huanfeng Chemical Machinery Trial Plant, Beijing, China). 

#### 2.4.6. Mechanical Properties

The tensile mechanical properties of the PDEBEG/CB compounds were investigated on the dumbbell specimens (ca. 25 × 6 × 2 mm) with a SANS CMT 4104 electrical tensile instrument at room temperature and a tensile rate of 500 mm/min according to ISO/DIS 37-1990. Hardness characterization was made by Rockwell hardness tester (Bareiss Prufgeratebau GmbH, Oberdischingen, Germany) using a 6 mm plate-shaped material in height gauge. In the measurement of permanent set, the fractured sample was placed for 3 min, the two parts of the fracture were fitted together, and the length between two parallel lines after the anastomosis was measured with a measuring instrument, which was calculated by the Equation:$$E_{b}=\frac{100\left(L_{b}-L_{0}\right)}{L_{0}}$$
where *E_b_* stands for the elongation at break, *L_b_* stands for the gauge length at the time of sample fracture, and *L*_0_ stands for the initial gauge length of the sample.

#### 2.4.7. Rubber Process Analyzer (RPA)

Strain sweep experiments (storage modulus (G’) as a function of scanning strain) were performed on vulcanizates by an RPA2000 rubber process analyzer (Alpha Technologies Corporation, Akron, OH, USA) at 60 °C.

#### 2.4.8. Scanning Electron Microscopy (SEM)

The morphology of the blends was determined by scanning electron microscopy (SEM, S4800 Hitachi Co., Tokyo, Japan) at 10 kV. After immersing in liquid nitrogen for 5 min, notched samples were fractured by vice and then sprayed with a thin gold layer. 

#### 2.4.9. Oil Resistance Test

Oil resistance of vulcanizates was carried out according to GB/T 1690-2010. Tensile test pieces of the blends were immersed in ASTM oil 3# at 150 °C for 72 h. After completion of the aging, the test specimen was cooled to room temperature and blotted with filter paper. The mechanical properties and hardness of the aged samples were measured at 25 °C, as described above.

## 3. Results and Discussion

### 3.1. Synthesis and Characterization of the PDEBEG

Redox-initiated emulsion polymerization was carried out at 30 °C to synthesize the PDEBEG elastomer. We used 40 wt% butyl acrylate to ensure the low-temperature resistance and certain elasticity of the elastomer, and adjusted the ratio of diethyl itaconate to ethyl acrylate to optimize the mechanical properties and oil resistance of the elastomer. 

As shown in Table 3, the number average molecular weight of the PDEBEG is more than 200,000, the yield is up to 96%, and the molecular polydispersity index (*Đ*) is between 2.5 and 4.0. The more the amount of DEI introduced, the lower gel content of PDEBEG. The introduction of DEI is beneficial to reduce the gel content of the elastomer. It can be explained by the increase of the *Tg* and the decrease of the molecular weight with the increase of DEI content, which decreased the possibility of macromolecular entanglements.

The FT-IR spectra of the PDEBEG are displayed in Figure 3. The absorption peaks at 2951, 2924, and 2863 cm^−1^ are attributed to the stretching vibrations of –CH_3_, –CH_2_, and –CH of the PDEBEG, respectively. Characteristic peaks are found at 1734 cm^−1^ (C=O ester), while peaks at 1029 and 1168 cm^−1^ are attributed to the C–O–C symmetrical and antisymmetric stretching vibration of PDEBEG, respectively. The small peak at 911 cm^−1^ corresponds to the ring vibration of the epoxy group. The results indicate that GMA has been successfully introduced into the polymer chains. 

The chemical structures and compositions of the PDEBEG were further determined by using ^1^H NMR spectroscopy. Figure 4 shows the ^1^H NMR spectrum of PDEBEG-30. The signals at 3.68, 4.08, and 1.20 ppm correspond to the protons of the methylene and methyl groups in diethyl itaconate. Those at 3.97, 1.47, 1.19, and 0.90 ppm are assigned to the protons of the methylene and methyl groups in butyl acrylate of the PDEBEG chains. Those of the protons of the methylene and methyl groups in ethyl acrylate appear at 4.06 and 1.21 ppm, respectively. The small peaks at 3.82 ppm originate from the protons of an epoxy group’s adjacent methylene, indicating that GMA has been successfully introduced into the polymer chains.

### 3.2. Thermal Performance of the PDEBEG

The glass transition temperature (*Tg*) determines whether a material is an elastomer at the end-use temperature. In Figure 5, nonisothermal DSC results from −40 to 40 °C for PDEBEG samples exhibited a completely amorphous structure. *Tg* increases with the increase of the DEI content from −25.2 to −0.8 °C. The introduction of the ester groups is responsible for the increase of *Tg* because this polar group can increase the interaction between the molecular chains and restrict the movement of polymer chain segments. 

The thermal stability of the PDEBEG was investigated by thermogravimetric analysis (TGA). The TGA and TGA derivative curves of PDEBEG measured from 30 to 800 °C are shown in Figure 6, and the thermal properties of the PDEBEG are listed in Table 4. The initial decomposition temperature is about 300 °C. The 5% weight loss and the maximum decomposition temperatures range from 346–363 °C and 385–411 °C, respectively. With the introduction of the DEI, the thermal decomposition temperature of the PDEBEG gradually decreases, indicating that the ester groups in diethyl itaconate reduce the thermal stability of the PDEBEG.

### 3.3. Vulcanization Characteristics, Morphology, and Mechanical Properties of the PDEBEG/CB Composites

Most elastomer products are useless unless properly cured. Since the modulus of the elastomer increases dramatically during curing, it is used to monitor the curing progress. Table 5 shows the characteristic curing data for the PDEBEG with various DEI contents and AR72LS (prepared in the same way as the PDEBEG composites) at 180 °C. As the fraction of the DEI decreases, the torque difference increases and the optimum cure time (the time when the torque reaches 90% of the maximum torque) increases, indicating an increase in crosslink density. 

Figure 7 shows the strain amplitude dependence of G’ of the unvulcanized PDEBEG/CB compounds. In the filled elastomers, the structure of the filler–filler networks can be reflected by the Payne effect, which is referred to as the strain dependence of the G’. The G’ value decreases sharply when the filler–filler networks are broken after the strain reaches a critical value. The initial G’ of the PDEBEG/CB decreases with the increase of the DEI content due to the improvement of the CB dispersion. As the DEI content increases, the gel content of the PDEBEG decreases, resulting in weak filler–filler network and decreased initial G’ of the PDEBEG/CB. 

The dispersion of a filler in the matrix is an important factor for a composite to achieve high performance. The SEM micrographs shown in Figure 8 exhibit the dispersion of CB in the PDEBEG/CB. The light spots in the micrographs are the CB particles. As observed in Figure 8a–f, the filler dispersion in the PDEBEG matrix is obviously improved with the incorporation of the DEI, which is mutually confirmed by the RPA results.

The typical stress–strain curves of the PDEBEG/CB with various DEI content and AR72LS/CB are shown in Figure 9, and Table 6 summarizes the acquired data. The results show the tensile strength of PDEBEG/CB increases with the decrease of the DEI content. The small permanent set (Table 6) indicates that the PDEBEG elastomers have good recovery ability. For the PDEBEG-10/CB, the tensile strength can reach 14.5 MPa, the elongation at break is up to 305%, and the permanent deformation is only 10%, indicating that the overall mechanical properties of the PDEBEG-10/CB are superior to those of the commercially available acrylate rubber AR72LS.

### 3.4. Oil Resistance of the PDEBEG/CB Composites

Since the PDEBEG is expected to replace the commercially available acrylate rubber and be used as an oil seal material at a high temperature, its oil resistance is also an important parameter to be evaluated. According to the test standard of oil-resistant rubber GB/T 1690-2010, the oil resistance test for the PDEBEG/CB was carried out using ASTM 3# oil (150 °C × 72 h). As shown in Figure 10, we can clearly compare the mechanical performance of the PDEBEG/CB with that of AR72LS/CB before and after soaking. Table 7 shows the retention of tensile strength, elongation at break, and hardness of the PDEBEG, which are reduced after soaking. Both the quality and volume of the PDEBEG increased after soaking because of the swelling of the elastomer network. The above Figure 10 and Table 7 show that the mechanical properties and oil resistance of the PDEBEG/CB with 10–20 wt% DEI are comparable to those of the commercially available acrylate AR72LS. The content of ester groups and the cross-linking density are the main factors affecting oil resistance. Molecular chains with polar ester groups can contribute to oil resistance, while molecular networks with high cross-linking density can inhibit the penetration of oil. The presence of DEI units affects both the ester group content and the cross-linking density of the PDEBEG. The presence of a diester group in the DEI structure makes the PDEBEG a high ester group content, which is advantageous to the oil resistance of the PDEBEG. However, the cross-linking density is lowered due to the steric hindrance effect of the diester groups, which is disadvantageous to the oil resistance of the PDEBEG.

## 4. Conclusions

Novel oil-resistant elastomers (PDEBEG) with bio-based diethyl itaconate, butyl acrylate and ethyl acrylate as main monomers were designed and synthesized by redox emulsion polymerization. As the content of diethyl itaconate increased, the gel content obviously decreased and the dispersion of filler improved greatly, while the thermal stability of the PDEBEG gradually reduced. For the PDEBEG with 10 wt% diethyl itaconate, the number average molecular weight is 497,000, the conversion rate is over 96%, the glass transition temperature is −22.7 °C, the tensile strength is up to 14.5 MPa, and the elongation at break can reach 305%. The PDEBEG/CB composites were subjected to a high-temperature oil test at 150 °C for 72 h. The results show that high-temperature oil resistance of PDEBEG-10 and PDEBEG-20 is superior to that of the commercially available acrylate rubber AR72LS. Overall, the novel PDEBEG is a promising candidate for the next generation of oil-resistant rubber.

## Figures and Tables

**Figure 1 polymers-11-01897-f001:**
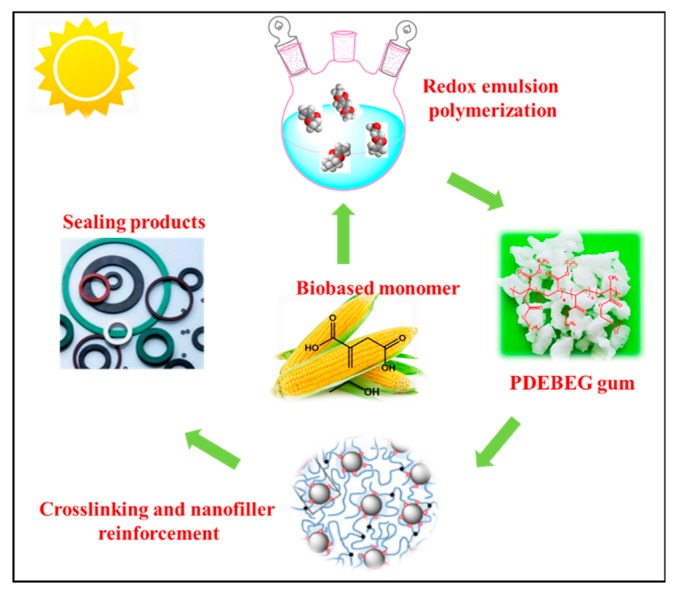
Design and preparation route of the novel elastomer with bio-based diethyl itaconate.

**Figure 2 polymers-11-01897-f002:**
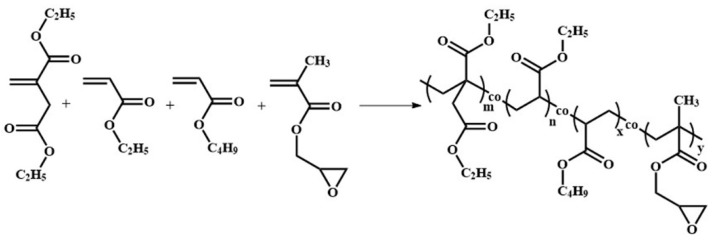
Reaction formula of PDEBEG by redox-initiated emulsion polymerization.

**Figure 3 polymers-11-01897-f003:**
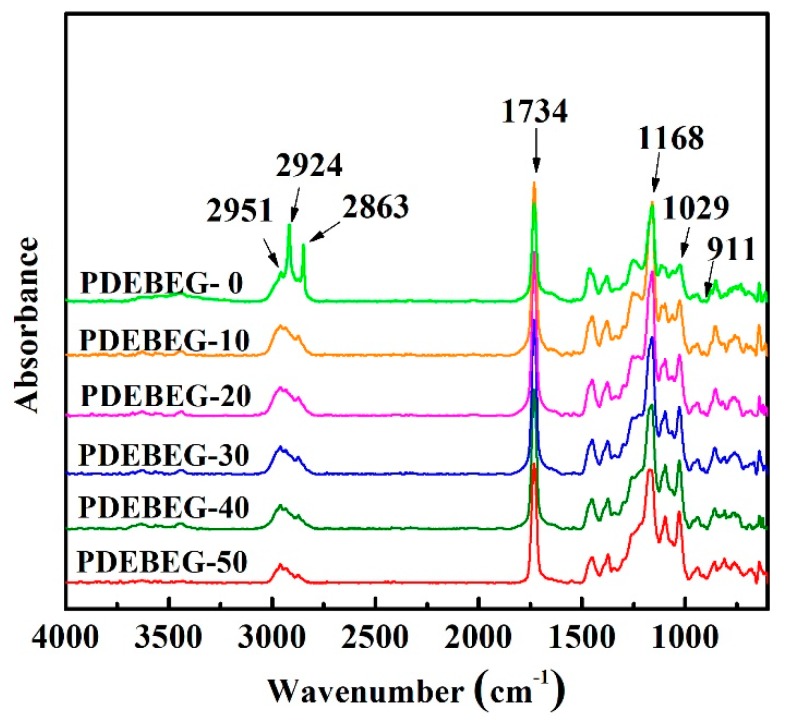
FT-IR spectra of PDEBEG.

**Figure 4 polymers-11-01897-f004:**
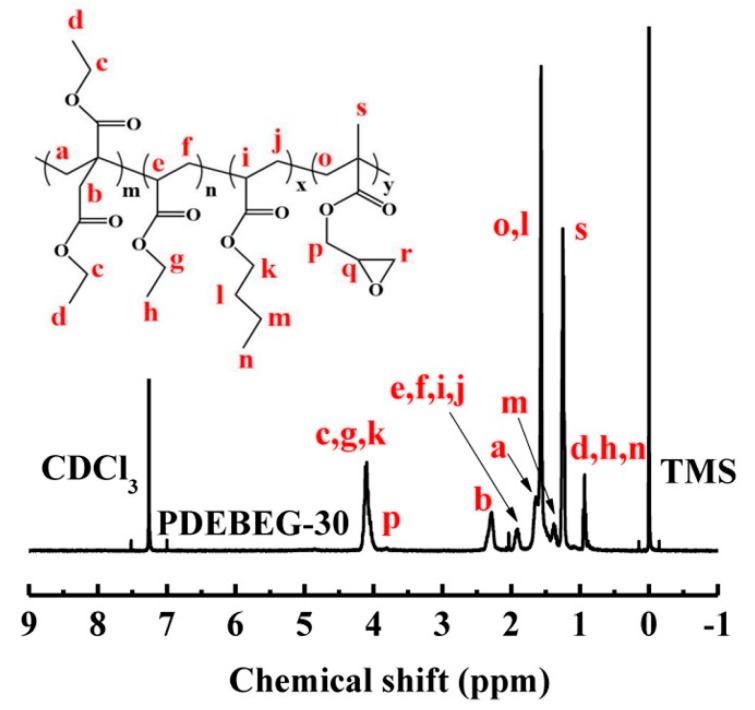
^1^H NMR spectra of PDEBEG-30.

**Figure 5 polymers-11-01897-f005:**
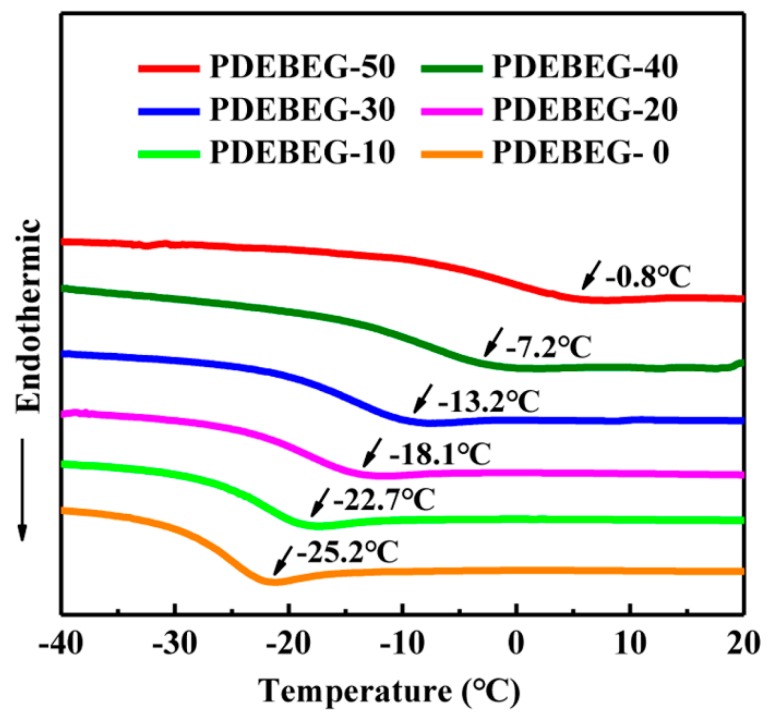
DSC thermograms of PDEBEG with different diethyl itaconate contents.

**Figure 6 polymers-11-01897-f006:**
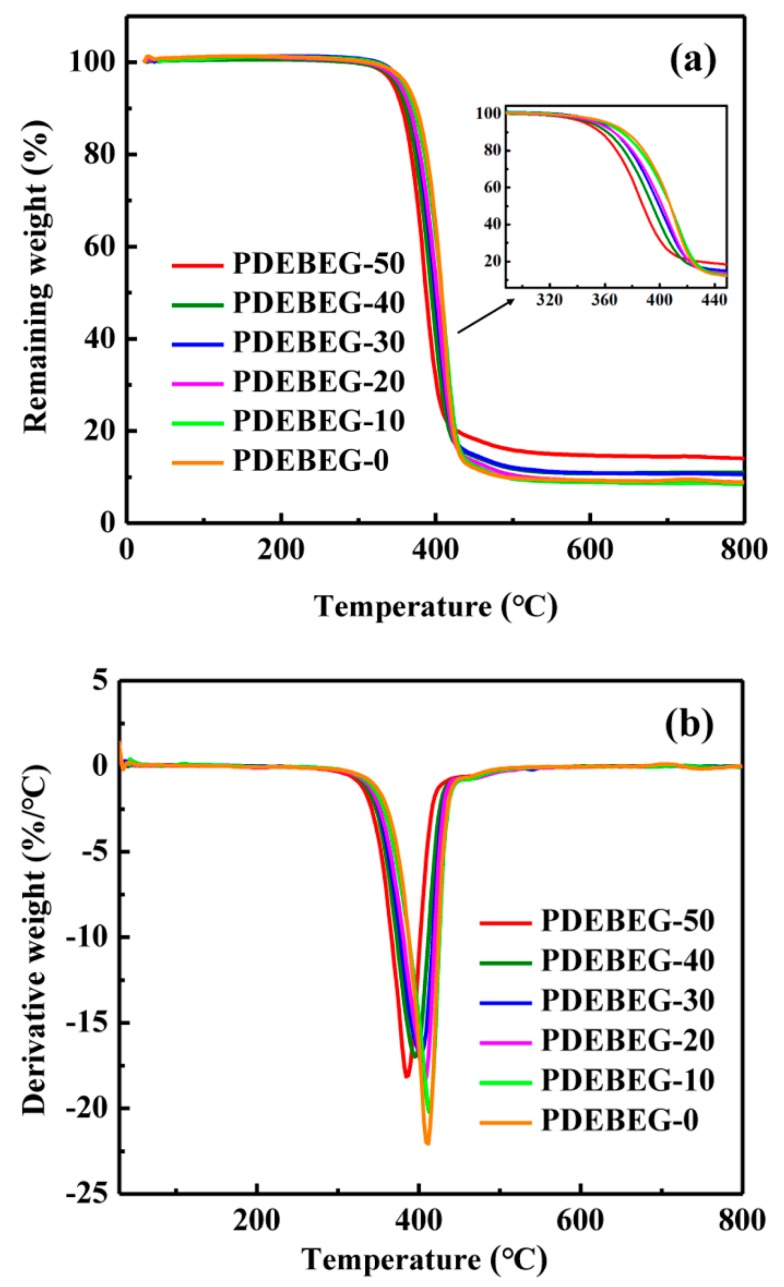
(**a**) TGA and (**b**) TGA derivative curves of PDEBEG.

**Figure 7 polymers-11-01897-f007:**
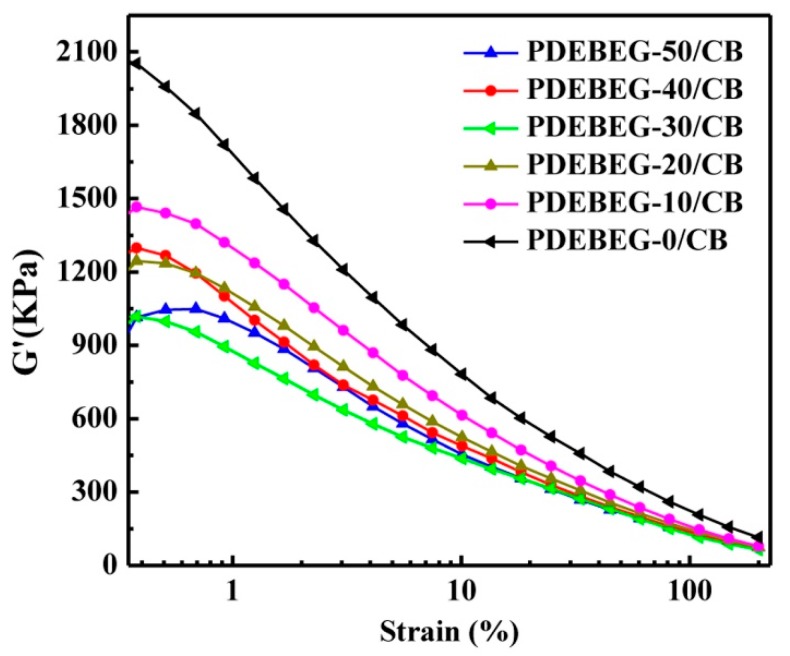
Strain amplitude dependence of G’ of the PDEBEG/CB compounds.

**Figure 8 polymers-11-01897-f008:**
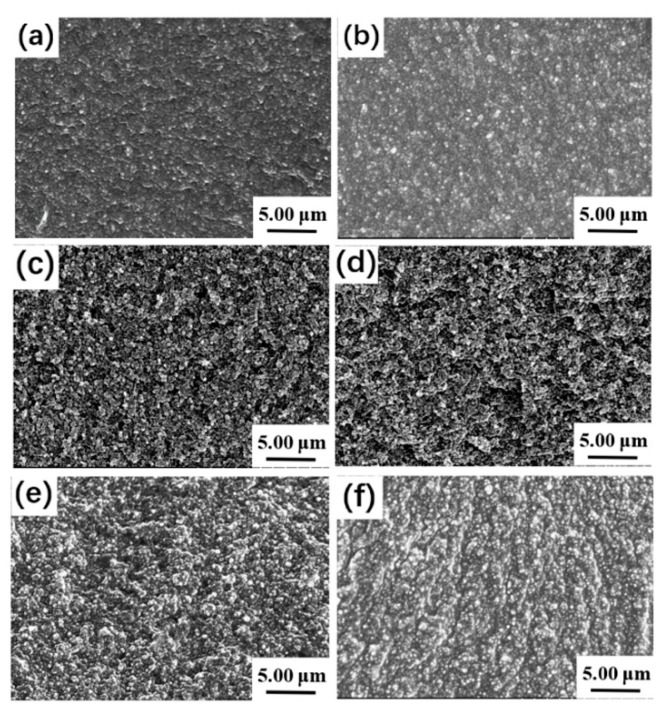
Scanning electron microscope graphs of (**a**) PDEBEG-50/CB, (**b**) PDEBEG-40/CB, (**c**) PDEBEG-30/CB, (**d**) PDEBEG-20/CB, (**e**) PDEBEG-10/CB, (**f**) PDEBEG-0/CB.

**Figure 9 polymers-11-01897-f009:**
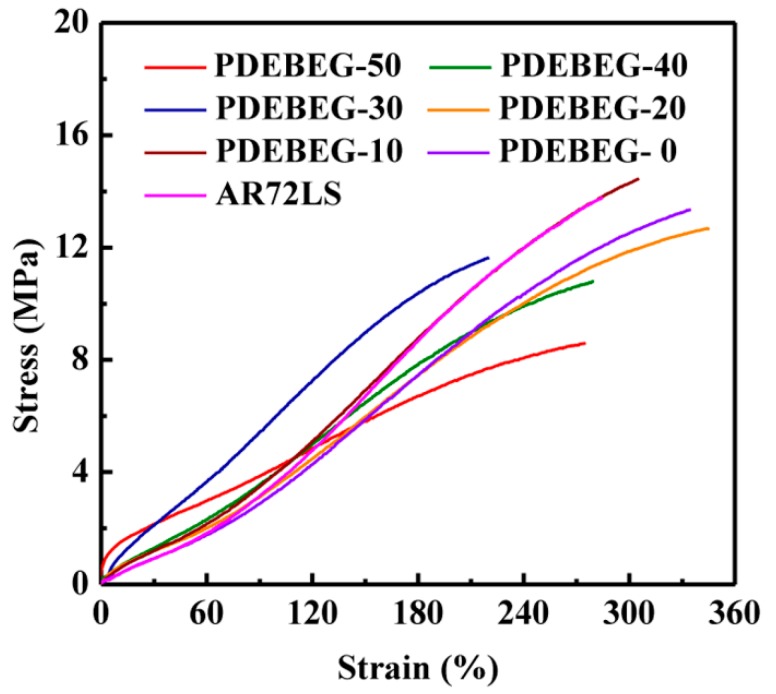
Stress–strain curves of the PDEBEG/CB and AR72LS/CB nanocomposites.

**Figure 10 polymers-11-01897-f010:**
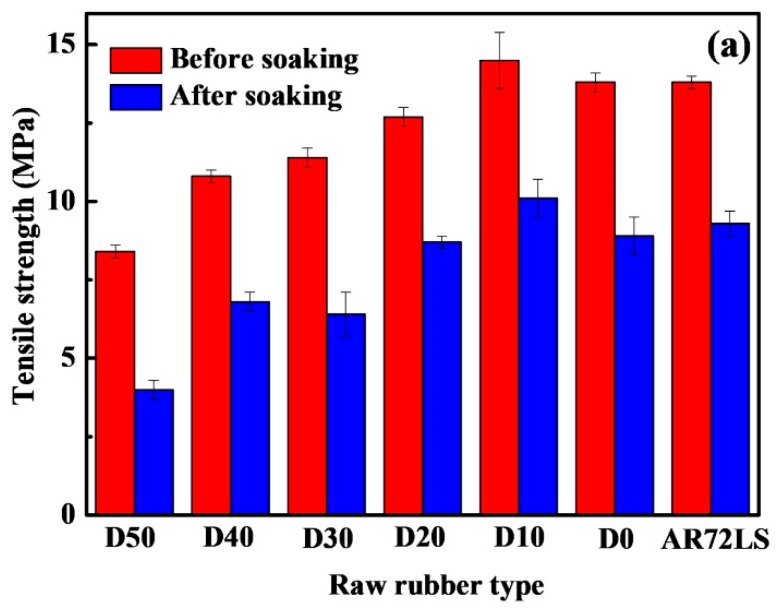
Mechanical performance comparison histogram before and after soaking, (**a**) tensile strength, (**b**) elongation at break. (The D50, D40, D30, D20, D10 and D0 refer to PDEBEG-50/CB, PDEBEG-40/CB, PDEBEG-30/CB, PDEBEG-20/CB, PDEBEG-10/CB, PDEBEG-0/CB, respectively.)

**Table 1 polymers-11-01897-t001:** Mass ratio of monomers for redox-initiated emulsion polymerization of PDEBEG.

Ingredients	Diethyl Itaconate (wt%)	Ethyl Acrylate (wt%)	Butyl Acrylate (wt%)	Glycidyl Methacrylate (wt%)
PDEBEG-50 ^a^	50	10	40	2
PDEBEG-40	40	20	40	2
PDEBEG-30	30	30	40	2
PDEBEG-20	20	40	40	2
PDEBEG-10	10	50	40	2
PDEBEG-0	0	60	40	2

^a^ The feed fraction of the diethyl itaconate, ethyl acrylate, and butyl acrylate for the PDEBEG was 50 wt%, 10 wt%, and 40 wt%, respectively, while 2 wt% of glycidyl methacrylate was introduced. The number (50) in the PDEBEG-50 indicates the proportion of diethyl itaconate (bio-based content) in the feeding process. The rest is the same as above.

**Table 2 polymers-11-01897-t002:** Recipe of the PDEBEG/CB composites.

Ingredients	Loading (phr) ^a^
PDEBEG	100.0
N330	60.0
stearic acid	1.0
antioxidant 445	1.0
zinc dibutyl dithiocaarbamate	2.5
Sulfur	1.0

^a^ phr is the part per hundred of rubber.

**Table 3 polymers-11-01897-t003:** Molecular weight, yield, and composition of redox-initiated PDEBEG.

Sample	*M*_n_/10^4^	*Đ* ^a^	Yield/%	Gel Content/%
PDEBEG-50	23.7	3.86	96.1	7
PDEBEG-40	30.6	3.54	97.5	12
PDEBEG-30	31.9	3.41	98.2	14
PDEBEG-20	51.7	3.03	97.8	23
PDEBEG-10	49.7	3.61	97.6	38
PDEBEG-0	70.2	2.84	98.2	46

^a^*Đ* stands for polydispersity index (Mw/Mn).

**Table 4 polymers-11-01897-t004:** Thermal properties of PDEBEG.

Samples	PDEBEG-50	PDEBEG-40	PDEBEG-30	PDEBEG-20	PDEBEG-10	PDEBEG-0
T_d,5%_ ^a^	346	350	357	354	360	363
T_d,max_ ^b^	385	395	403	407	413	416

^a^ Temperature at which a 5% weight loss was observed in the TGA traces recorded at 10 °C/min. ^b^ Temperature of maximum degradation rate.

**Table 5 polymers-11-01897-t005:** Curing parameters for the PDEBEG/CB and AR72LS/CB elastomers.

Sample	Scorch Time (min:s)	Curing Time (min:s)	Torque Increase (dNm)
PDEBEG-50	0:40	13:24	10.3
PDEBEG-40	0:11	9:46	14.0
PDEBEG-30	1:23	17:23	12.7
PDEBEG-20	1:04	15:23	12.5
PDEBEG-10	0:16	14:23	20.0
PDEBEG-0	1:46	21.50	18.5
AR72LS	3:03	19:17	19.0

**Table 6 polymers-11-01897-t006:** Mechanical properties of the cross-linked PDEBEG/CB and AR72LS/CB elastomers.

Sample	Tensile Strength (MPa)	Elongation at Break (%)	Permanent Set (%)	Hardness (Shore A)
PDEBEG-50	8.4 ± 0.2	249 ± 13	8 ± 2	72 ± 1
PDEBEG-40	10.8 ± 0.2	280 ± 10	8 ± 1	69 ± 1
PDEBEG-30	11.4 ± 0.3	234 ± 14	6 ± 2	76 ± 2
PDEBEG-20	12.7 ± 0.3	345 ± 27	12 ± 2	63 ± 1
PDEBEG-10	14.5 ± 0.4	305 ± 23	10 ± 1	63 ± 1
PDEBEG-0	13.8 ± 0.3	346 ± 25	12 ± 1	60 ± 1
AR72LS	13.8 ± 0.2	285 ± 6	12 ± 2	66 ± 1

**Table 7 polymers-11-01897-t007:** Retention of mechanical properties of PDEBEG/CB and AR72LS/CB vulcanizates before and after soaking in ASTM 3# oil at 150 °C for 72 h.

Sample	Retention Rate of Tensile Strength (%)	Retention Rate of Elongation at Break (%)	Retention Rate of Shore A Hardness (%)	Δm (%)	ΔV (%)
PDEBEG-50	47.6	80.3	72.2	+19.6	+23.7
PDEBEG-40	63.0	80.4	75.4	+18.5	+17.6
PDEBEG-30	56.1	82.9	69.7	+19.2	+18.8
PDEBEG-20	68.5	92.8	88.9	+16.9	+12.5
PDEBEG-10	69.7	89.2	92.1	+13.8	+9.9
PDEBEG-0	64.5	64.5	86.7	+18.5	+16.7
AR72LS	67.4	79.6	86.4	+18.0	+14.0

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
