# Peer review of "Design, Preparation, and Evaluation of a Novel Elastomer with Bio-Based Diethyl Itaconate Aiming at High-Temperature Oil Resistance"

_polymers, 2019, doi:10.3390/polym11111897_

Round 1

Reviewer 1 Report

The study put forward by Hui Yang et al., presents a new solution for synthesis a novel elastomer - poly(diethyl itaconate/butyl acrylate/ethyl acrylate/glycidyl methacrylate) (PDEBEG) via redox emulsion polymerization based on bio-based diethyl itaconate, butyl acrylate, ethyl acrylate and glycidyl methacrylate. In addition, the authors demonstrated the impact of the method used on the characteristics of the polymer itself as well as on the properties of carbon black reinforced composites.

The paper reports on an interesting issue in which important contributions are being performed recently. In general the paper is of a above average value with respect to broadening knowledge and extending methods of elastomer characterization. Both the part regarding synthesis method and the characteristics of finished materials has been thoroughly presented and described. Large number of tests carried out, including various research methods and an in-depth description of the results, highlights the high value of the article.

The way the results are presented is clear and legible. Figures and tables understandable. The description of the results forms a unified whole creating a logical cause and effect sequence. Stylistics of language at a satisfactory level.

I recommend the paper to be publish in the Polymers journal after making minor revision. Below you can find my small comments concerning this article. I hope they will be helpful for improving the article:

In the "Materials and methods" section, please add descriptions to the following tests that were carried out: Vulcanizattion characteristisc Hardness Permanent set Please add the standard deviation for mechanical properties (TS, EB, permanent set, hardness) - tables 6 and 7 and figure 10. The choice of sample size for mechanical tests is incomprehensible to me (P. 5, L. 141). According to the ASTM D412 standard, we have a choice of Type A, B, C, D, E and F, but none of them match the dimensions presented in the manuscript. Please comment.

Author Response

The study put forward by Hui Yang et al., presents a new solution for synthesis a novel elastomer - poly(diethyl itaconate/butyl acrylate/ethyl acrylate/glycidyl methacrylate) (PDEBEG) via redox emulsion polymerization based on bio-based diethyl itaconate, butyl acrylate, ethyl acrylate and glycidyl methacrylate. In addition, the authors demonstrated the impact of the method used on the characteristics of the polymer itself as well as on the properties of carbon black reinforced composites.

The paper reports on an interesting issue in which important contributions are being performed recently. In general the paper is of a above average value with respect to broadening knowledge and extending methods of elastomer characterization. Both the part regarding synthesis method and the characteristics of finished materials has been thoroughly presented and described. Large number of tests carried out, including various research methods and an in-depth description of the results, highlights the high value of the article.

The way the results are presented is clear and legible. Figures and tables understandable. The description of the results forms a unified whole creating a logical cause and effect sequence. Stylistics of language at a satisfactory level.

I recommend the paper to be publish in the Polymers journal after making minor revision. Below you can find my small comments concerning this article. I hope they will be helpful for improving the article:

In the "Materials and methods" section, please add descriptions to the following tests that were carried out: Vulcanizattion characteristisc Hardness Permanent set Please add the standard deviation for mechanical properties (TS, EB, permanent set, hardness) - tables 6 and 7 and figure 10. The choice of sample size for mechanical tests is incomprehensible to me (P. 5, L. 141). According to the ASTM D412 standard, we have a choice of Type A, B, C, D, E and F, but none of them match the dimensions presented in the manuscript. Please comment.

Answer: Thanks so much for the comment. We have added descriptions of the curing characteristisc, hardness and permanent set in the "Measurements and Characterization". In addition, we have added the standard deviation for mechanical properties in the table 6 and figure 10. Table 7 has been revised according to the comments of reviewer #2. Furthermore, we have checked carefully the sample size for mechanical tests and corrected it in the revised manuscript, according to ISO/DIS 37-1990 standard, we had a choice of Type 1.

2.4.5 Curing characteristics

The curing characteristics of the PDEBEG/CB compounds were measured at 180℃ using a P3555B2 disc vulkameter (Beijing Huanfeng Chemical Machinery Trial Plant, Beijing, China).

2.4.6 Mechanical properties

The tensile tests were conducted according to ASTM D412. All the specimens were dumbbell-shaped with a gauge length of 25 mm, a width of 6 mm, and a thickness of 2 mm. The specimens were tested on a CMT4104 electronic tensile tester (SANS, China). At least five specimens of each composite were tested for an average value.

The tensile mechanical properties of the PDEBEG/CB compounds were investigated on the dumbbell specimens (ca. 25 mm × 6 mm ×2 mm) with a SANS CMT 4104 electrical tensile instrument at room temperature and a tensile rate of 500 mm/min according to ISO/DIS 37-1990. Hardness characterization was made by Rockwell hardness tester (Bareiss Prufgeratebau GmbH, Oberdischingen, Germany) using a 6 mm plate-shaped material in height gauge. In the measurement of permanent set, the fractured sample was placed for 3 min, and the two parts of the fracture were fitted together, and the length between two parallel lines after the anastomosis was measured with a measuring instrument, which was calculated by the equation:

Eb stands for the elongation at break, Lb stands for the gauge length at the time of sample fracture, and L0 stands for the initial gauge length of the sample.

Table 6. Mechanical properties of the cross-linked PDEBEG/CB and AR72LS/CB elastomers.

Sample

Tensile strength

(MPa)

Elongation at break

(%)

Permanent set

(%)

Hardness

(Shore A)

PDEBEG-50

8.4 ± 0.2

249 ± 13

8 ± 2

72 ± 1

PDEBEG-40

10.8 ± 0.2

280 ± 10

8 ± 1

69 ± 1

PDEBEG-30

11.4 ± 0.3

234 ± 14

6 ± 2

76 ± 2

PDEBEG-20

12.7 ± 0.3

345 ± 27

12 ± 2

63 ± 1

PDEBEG-10

14.5 ± 0.4

305 ± 23

10 ± 1

63 ± 1

PDEBEG-0

13.8 ± 0.3

346 ± 25

12 ± 1

60 ± 1

AR72LS

13.8 ± 0.2

285 ± 6

12 ± 2

66 ± 1

Figure 10. Mechanical performance comparison histogram before and after soaking, (a) tensile strength, (b) elongation at break. (The D50, D40, D30, D20, D10 and D0 refer to the PDEBEG-50/CB, PDEBEG-40/CB, PDEBEG-30/CB, PDEBEG-20/CB, PDEBEG-10/CB, PDEBEG-0/CB, respectively.).

Table 7. Retention of mechanical properties of PDEBEG/CB and AR72LS/CB vulcanizates before and after soaking in ASTM 3# oil at 150℃ for 72 h.

Sample

Retention rate of tensile strength (%)

Retention rate of

elongation at break (%)

Retention rate of shore A hardness (%)

Δm%

ΔV%

PDEBEG-50

47.6

80.3

72.2

+19.6

+23.7

PDEBEG-40

63.0

80.4

75.4

+18.5

+17.6

PDEBEG-30

56.1

82.9

69.7

+19.2

+18.8

PDEBEG-20

68.5

92.8

88.9

+16.9

+12.5

PDEBEG-10

69.7

89.2

92.1

+13.8

+9.9

PDEBEG-0

64.5

64.5

86.7

+18.5

+16.7

AR72LS

67.4

79.6

86.4

+18.0

+14.0

Reviewer 2 Report

The manuscript "Design, Preparation and Evaluation of a Novel Elastomer with Bio-Based Diethyl Itaconate Aiming at High-Temperature Oil Resistance " by Wang and coworkers deals with the formation of elastomirc polymers for carbon black composites. It shows some interesting results. Nevertheless, I have several comments.

General comments:

1. In my opinion the manuscript shows a lot of data with a brief description. There should be much more discussion and also comments about the reasons for the observations.

2. The polymerization method emulsion polymerization should be mentioned in the introduction as well. Is there anything about green emulsion polymerization?

https://www.sciencedirect.com/science/article/pii/S0300944005000196

https://pubs.rsc.org/en/content/articlelanding/2017/py/c7py00464h/unauth#!divAbstract

https://www.sciencedirect.com/science/article/pii/S0032386117301593

https://patents.google.com/patent/US9932421B2/en

https://pubs.acs.org/doi/abs/10.1021/acs.langmuir.8b01708

3. The progress of Tg in Figure 5 should be described in the text, commented and discussed.

4. Is there any information about the copolymerization of itaconate esters with the used monomers? Copolymerisation parameters etc.?

5. There should be more discussion on the mechanical properties.

6. How does the crosslinking proceed in the present case?

Specific comments:

7. The expression "relatively" in the abstract is very vague, the range of tuneable glass transitions temperatures should be mentioned as well.

8. What are "oil-based places"? The autors should be more specific.

9. The expression "cross-linkable ability" is not really good.

10. Also "low temperature resistance" is not a good expression. What is resistant?

11. Experimental: "coagulated to form a coagulate" does not really make sense

12. Experimental: I would suggest to remove the constant parts from Table 1 into the text and make a table with all variables for the monomers

13. Between the repeating units in Figure 1 should be "co". At the moment it looks like a block copolymer

14. The yield in Table 3 should have a digit less.

15. The % do not add up in the footnote of Table 3

16. Instead of Mw/Mn, italic "Đ" should be used

17. In Figure 4 some huge peaks are not assigned

18. In my opinion some of the tables can combined. The same goes for some of the Figures.

19. In Figure 7 "CB" should be added to the legend.

20. The journal abbreviations should be checked.

Author Response

The manuscript "Design, Preparation and Evaluation of a Novel Elastomer with Bio-Based Diethyl Itaconate Aiming at High-Temperature Oil Resistance " by Wang and coworkers deals with the formation of elastomirc polymers for carbon black composites. It shows some interesting results. Nevertheless, I have several comments.

General comments

(1) In my opinion the manuscript shows a lot of data with a brief description. There should be much more discussion and also comments about the reasons for the observations.

Answer: Thanks so much for the comment. We have added some discussions and comments for our manuscript’s data, such as thermal Performance, mechanical properties, oil resistance etc. You can find these discussions in the following responses.

(2) The polymerization method emulsion polymerization should be mentioned in the introduction as well. Is there anything about green emulsion polymerization?

https://www.sciencedirect.com/science/article/pii/S0300944005000196

https://pubs.rsc.org/en/content/articlelanding/2017/py/c7py00464h/unauth#!divAbstract

https://www.sciencedirect.com/science/article/pii/S0032386117301593

https://patents.google.com/patent/US9932421B2/en

https://pubs.acs.org/doi/abs/10.1021/acs.langmuir.8b01708

Answer: Thanks so much for the comments. Related information has been added in the revised manuscript, see below:

More commonly, dialkyl itaconate is used as a substitute for itaconic acid to homopolymerize or copolymerize with other unsaturated monomers by emulsion polymerization [24, 36-38]. This emulsion polymerization which an isolated latex particle is formed by a micelle mechanism in an aqueous medium-based emulsion and a free polymerization reaction is carried out therein to produce a high polymer, is a green polymerization method. Nowadays, it includes surface-functionalized latexes [39], using a surfactant and organic solvent free emulsion system [40], using lignin-based polymeric surfactants [41] and Disponil AFX non-ionic surfactants [42], which has been used widely by green emulsion polymerization.

Durant, Y.; Jiang, B.; Tsavalas, J. Emilsion polymerization of ester of itaconic acid. U. S. Pat. 2018, 932, 421. Chakraborty, S.; Ramakrishnan, S. Surface-Functionalized Polystyrene Latexes Using Itaconate-Based Surfmers. Langmuir 2018, 34, 11729−11737. Qiao, Z.; Qiu, T.; Liu, W.; Zhang, L.; Tu, J.; Guo, L.; Li, X. A “green” method for preparing ABCBA pentablock elastomers by using RAFT emulsion polymerization. Polym. Chem. 2017, 8, 3013. Schmidt, B.; Molinari, V.; Esposito, D.; Tauer, K.; Antonietti, M. Lignin-based polymeric surfactants for emulsion polymerization. Polymer 2017, 112, 418-426. Fernandez, A.M.; Held, U.; Willing, A.; Breuer, W.H. New green surfactants for emulsion polymerization. Prog. Org. Coat. 2005, 53, 246-255.

(3) The progress of Tg in Figure 5 should be described in the text, commented and discussed.

Answer: Thanks so much for the comment. We have added some discussions and comments for figure 5.

In Figure 5, nonisothermal DSC results from -40 to 40°C for PDEBEG samples exhibited a completely amorphous structure. Tg increases with the increase of the DEI content from -25.2 to -0.8°C. The introduction of the ester groups is responsible for the increase of Tg because this polar group can increase the interaction between the molecular chains and restrict the movement of polymer chain segments.

(4) Is there any information about the copolymerization of itaconate esters with the used monomers? Copolymerisation parameters etc.?

Answer: Thanks so much for the comment. It is a big challenge to obtain the copolymerization paramters etc. because it is a quaternary polymerization system, and we need to conduct lots of experiments to get the corresponding parameters. We will explore this problem in the future research.

(5) There should be more discussion on the mechanical properties.

Answer: Thanks so much for the comment. We have added some discussions and comments on the mechanical properties.

The typical stress-strain curves of the PDEBEG/CB with various DEI content are shown in Figure 9, and Table 6 summarizes the acquired data. The results show the tensile strength of PDEBEG/CB increases with the decrease of the DEI content. The small permanent set (Table 6) indicates that the PDEBEG elastomers have good recovery ability. For the PDEBEG-10/CB, the tensile strength can reach 14.5 MPa, the elongation at break is up to 305%, and the permanent deformation is only 10%, indicating that the overall mechanical properties of the PDEBEG-10/CB is superior to those of the commercially available acrylate rubber AR72LS.

(6) How does the crosslinking proceed in the present case?

Answer: Thanks so much for the comment. In the present case, we used sulfur as a promoter and zinc dibutyl dithiocaarbamate as a vulcanizing agent for crosslinking, which is consistent with the crosslinking mechanism the following literatures reported.

Yan, X.; Ji, Y.; He, T. Synthesis of fiber crosslinking cationic latex and its effect on surface properties of paper. Prog. Org. Coat. 2013, 76, 11-16.

Park, J.; Jana, S.C. Effect of plasticization of epoxy networks by organic modifier on exfoliztion of nanoclay. Macromolecules 2003, 36, 8391-8397.

Specific comments:

(7) The expression "relatively" in the abstract is very vague, the range of tuneable glass transitions temperatures should be mentioned as well.

Answer: Thanks so much for the comment. We are sorry for the puzzling statement in the original manuscript.

Page 1, Line 18: The PDEBEG has a number average molecular weight of more than 200,000 and a relatively high yield. the yield is up to 96%.

Page 1, Line 19: It is easy to control the glass transition temperature of the PDEBEG, which is ranged from -25.2 to -0.8℃ by adjusting the monomer ratio.

(8) What are "oil-based places"? The autors should be more specific.

Answer: Thanks so much for the comment. We are sorry for the puzzling statement in the original manuscript. We have corrected the description about "oil-based places".

Page 1, Line 30: It is widely used in oil-based places such as automobile manufacturing, aviation, and oil refining.

It is widely used to manufacture oil-resistant products in the automobile manufacturing, aerospace and machinery industries, such as oil pipes, drums and gaskets.

(9) The expression "cross-linkable ability" is not really good.

Answer: Thanks so much for the comment. We are sorry for the inappropriate statements we made in the manuscript.

Page 2, Line 63-69: In addition, the introduction of isoprene endows the elastomer with cross-linkable ability and low-temperature resistance. provides the crosslink points and soften the macromolecule chains. In order to balance the high-temperature oil resistance, thermal-oxidative aging and low-temperature resistance good elasticity of the elastomer with cross-linkable ability the crosslink points, the monomer isoprene should be replaced by other chemicals. In previous studies, the homopolymer of ethyl acrylate exhibited an excellent high-temperature oil resistance while the homopolymer of butyl acrylate exhibited a low-temperature resistance good elasticity [1].

(10) Also "low temperature resistance" is not a good expression. What is resistant?

Answer: Thanks so much for the comment. We are sorry for the inappropriate statements we made in the manuscript. We have corrected the inappropriate statements based on the comment.

We want to reduce the glass transition temperature of the polymer while ensuring the high-temperature oil resistance, making the molecular chain flexibility to ensure its elasticity.

Page 2, Line 63-69: In addition, he introduction of isoprene endows the elastomer with cross-linkable ability and low-temperature resistance. provides the crosslink points and soften the macromolecule chains. In order to balance the high-temperature oil resistance, thermal-oxidative aging and low-temperature resistance good elasticity of the elastomer with cross-linkable ability the crosslink points, the monomer isoprene should be replaced by other chemicals. In previous studies, the homopolymer of ethyl acrylate exhibited an excellent high-temperature oil resistance while the homopolymer of butyl acrylate exhibited a low-temperature resistance good elasticity [1].

(11) Experimental: "coagulated to form a coagulate" does not really make sense.

Answer: Thanks so much for the comment. We are sorry for the inappropriate statements we made in the manuscript.

Page 3 Line 103: It was then coagulated by ethanol to obtain the wet PDEBEG elastomer.

(12) Experimental: I would suggest to remove the constant parts from Table 1 into the text and make a table with all variables for the monomers.

Answer: Thanks so much for the comment. We have removed the constant patrts from Table 1 into the Part 2.2. and resume a table with all variables for the monomers.

Page 3, Line 97-103: The synthesis of the PDEBEG was carried out using mass ratios of DEI/EA/BA/GMA showed in Table 1. Deionized water (250.0 g, provided by Beijing University of Chemical Technology, China), SDBS (3.2 g), Fe-EDTA solution (8.0 g, concentration of aqueous solution is 1.0 wt.%), SHS solution (4.0 g, concentration of aqueous solution is 10.0 wt.%), and a mixture of the comonomers (DEI, EA, BA and GMA) were added into a three-neck glass flask under nitrogen atmosphere, and then the emulsion system was stirred at 400 rpm for 1 h. Once the initiator TBH was injected into the flask, the stirring rate was reduced to 210 rpm. After 8 h, the polymerization reaction was terminated by HA (0.4 g), and the PDEBEG latex was obtained.

Table 1. Mass ratio of monomers for redox-initiated emulsion polymerization of PDEBEG.

Ingredients

diethyl itaconate (wt.%)

ethyl acrylate (wt.%)

butyl acrylate

(wt.%)

glycidyl methacrylate (wt.%)

PDEBEG-50 a

50

10

40

2

PDEBEG-40

40

20

40

2

PDEBEG-30

30

30

40

2

PDEBEG-20

20

40

40

2

PDEBEG-10

10

50

40

2

PDEBEG-0

0

60

40

2

aThe feed fraction of the diethyl itaconate, ethyl acrylate and butyl acrylate for the PDEBEG is 50 wt.%, 10 wt.% and 40 wt.%, respectively, while 2 wt.% of glycidyl methacrylate is introduced. The number (50) in the PDEBEG-50 indicates the proportion of diethyl itaconate (bio-based content) in the feeding process. The rest is the same as above.

(13) Between the repeating units in Figure 1 should be "co". At the moment it looks like a block copolymer.

Answer: Thanks so much for the comment. We are sorry for the inappropriate statements we made in the manuscript. We have corrected the inappropriate statements based on the comment.

Page 1, Line 15-17: A novel elastomer poly(diethyl itaconate/butyl acrylate/ethyl acrylate/glycidyl methacrylate)(diethyl itaconate-co-butyl acrylate-co-ethyl acrylate-co-glycidyl methacrylate) (PDEBEG) was designed and synthesized by redox emulsion polymerization based on bio-based diethyl itaconate, butyl acrylate, ethyl acrylate and glycidyl methacrylate.

Page 2, Line 73-75: In this work, a novel elastomer poly(diethyl itaconate/butyl acrylate/ethyl acrylate/glycidyl methacrylate) (diethyl itaconate-co-butyl acrylate-co-ethyl acrylate-co-glycidyl methacrylate) (PDEBEG) with bio-based content was synthesized by a redox emulsion polymerization.

(14) The yield in Table 3 should have a digit less.

Answer: Thanks so much for the comment. We have corrected a digit less of the yield in Table 3.

Table 3. Molecular weight, yield and composition of redox-initiated PDEBEG.

Sample

Mn /104

Đa

Yield /%

Gel content/%

PDEBEG-50

23.7

3.86

96.1

7

PDEBEG-40

30.6

3.54

97.5

12

PDEBEG-30

31.9

3.41

98.2

14

PDEBEG-20

51.7

3.03

97.8

23

PDEBEG-10

49.7

3.61

97.6

38

PDEBEG-0

70.2

2.84

98.2

46

(15) The % do not add up in the footnote of Table 3.

Answer: Thanks so much for the comment. Some of the tables have been combined, so the original Table 3 was revised and the footnote of Table 3 was removed to that of Table 1.

(16) Instead of Mw/Mn, italic "Đ" should be used.

Answer: Thanks so much for the comment. We have used italic "Đ" instead of Mw/Mn in the revised manuscript.

Table 3. Molecular weight, yield and composition of redox-initiated PDEBEG.

Sample

Mn /104

Đa

Yield /%

Gel content/%

PDEBEG-50 a

23.7

3.86

96.1

7

PDEBEG-40

30.6

3.54

97.5

12

PDEBEG-30

31.9

3.41

98.2

14

PDEBEG-20

51.7

3.03

97.8

23

PDEBEG-10

49.7

3.61

97.6

38

PDEBEG-0

70.2

2.84

98.2

46

a Đ stands for polydispersity index (Mw/Mn).

(17) In Figure 4 some huge peaks are not assigned.

Answer: Thanks so much for the comment. In order to make the expression clearer, we deleted the 1H NMR spectrum of PDEBEG-0 and checked the chemical shift of PDEBEG-30. The huge peak in 7-8 ppm is assigned to deuterium generation of chloroform (CDCl3).

The chemical structures and compositions of the PDEBEG were further determined by using 1H NMR spectroscopy. Figure 4 shows that the 1H NMR spectrum of PDEBEG-0 and PDEBEG-30.

Figure 4. 1H NMR spectra of PDEBEG-30.

(18) In my opinion some of the tables can combined. The same goes for some of the Figures.

Answer: Thanks so much for the comment. We have combined some of the tables and the figures.

Table 6. Mechanical properties of the cross-linked PDEBEG/CB and AR72LS/CB elastomers.

Sample

Tensile strength

(MPa)

Elongation at break

(%)

Permanent set

(%)

Hardness

(Shore A)

PDEBEG-50

8.4 ± 0.2

249 ± 13

8 ± 2

72 ± 1

PDEBEG-40

10.8 ± 0.2

280 ± 10

8 ± 1

69 ± 1

PDEBEG-30

11.4 ± 0.3

234 ± 14

6 ± 2

76 ± 2

PDEBEG-20

12.7 ± 0.3

345 ± 27

12 ± 2

63 ± 1

PDEBEG-10

14.5 ± 0.4

305 ± 23

10 ± 1

63 ± 1

PDEBEG-0

13.8 ± 0.3

346 ± 25

12 ± 1

60 ± 1

AR72LS

13.8 ± 0.2

285 ± 6

12 ± 2

66 ± 1

Figure 10. Mechanical performance comparison histogram before and after soaking, (a) tensile strength, (b) elongation at break. (The D50, D40, D30, D20, D10 and D0 refer to the PDEBEG-50/CB, PDEBEG-40/CB, PDEBEG-30/CB, PDEBEG-20/CB, PDEBEG-10/CB, PDEBEG-0/CB, respectively.).

Table 7. Retention of mechanical properties of PDEBEG/CB and AR72LS/CB vulcanizates before and after soaking in ASTM 3# oil at 150℃ for 72 h.

Sample

Retention rate of tensile strength (%)

Retention rate of

elongation at break (%)

Retention rate of shore A hardness (%)

Δm%

ΔV%

PDEBEG-50

47.6

80.3

72.2

+19.6

+23.7

PDEBEG-40

63.0

80.4

75.4

+18.5

+17.6

PDEBEG-30

56.1

82.9

69.7

+19.2

+18.8

PDEBEG-20

68.5

92.8

88.9

+16.9

+12.5

PDEBEG-10

69.7

89.2

92.1

+13.8

+9.9

PDEBEG-0

64.5

64.5

86.7

+18.5

+16.7

AR72LS

67.4

79.6

86.4

+18.0

+14.0

(19) In Figure 7 "CB" should be added to the legend.

Answer: Thanks so much for the comment. Related information has been added in the Figure 7.

(20) The journal abbreviations should be checked.

Answer: Thanks so much for the comment. We have checked and corrected the journal abbreviations based on the comment.

Reference

Patil, A.O.; Coolbaugh, T.S. Elastomers: a literature review with emphasis on oil resistance. Rubber Chem. Technol. 2005, 78, 516-535. Liu, X.; Zhao, J.; Yang, R.; Lervolino R.; Barbera S. Effect of lubricating oil on thermal aging of nitrile rubber. Polym. Degrad. Stab. 2018, 151, 136-143. Zhao, X.; Xiang, P.; Tian, M.; Fong, H.; Jin, R.; Zhang, L. Nitrile butadiene rubber/hindered phenol nanocomposites with improved strength and high damping performance. Polymer 2007, 48, 6056-6063. Rajasedkar, R.; Pal, K.; Heinrich, G.; Das, A.; Das, C.K. Development of nitrile butadiene rubber-nanoclay composites with epoxidized natural rubber as compatibilizer. Mater. Des. 2009, 30, 3939-3845. Zhang, X.; Li, H.; Tian, D.; He, X.; Lu, C.H. Enhancement of thermal aging performance and oil resistance of acrylic rubber vulcanisates by adding devulcanised ground fluoroelastomer ultrafine powder as functional filler. Mater. Res. Innovations 2012, 16, 143-149. Wen, X.; Yuan, X.; Lan, L.; Hao, L.; Li, S.; Zheng, Z. Effect of transformer oil on room temperature vulcanized silicone rubber. IEEE Trans. Dielectr. Electr. Insul. 2017, 24, 2337-2343. Wang, Y.; Bi, L.; Zhang, H.; Zhu, X.; Liu, G.; Qiu, G.; Liu, S. Predictive power in oil resistance of fluororubber and fluorosilicone rubbers based on three-dimensional solubility parameter theory. Polym. Test. 2019, 75, 380-386. Ismail, S.; Chatterjee, T.; Naskar, K. Superior heat-resistant and oil-resistant blends based on dynamically vulcanized hydrogenated acrylonitrile butadiene rubber and polyamide 12. Polym. Adv. Technol. 2017, 28 665-678. Yeo, Y.G.; Park, H.H.; Lee, C.S. A study on the characteristics of a rubber blend of fluorocarbon rubber and hydrogenated nitrile rubber. J. Ind. Eng. Chem. 2013, 19, 1540-1548. Wang, J.; Li, H.; Zhang, L.; Lai, X.; Wu, W.; Zeng, X. In situ preparation of reduced graphene oxide reinforced acrylic rubber by self-assembly. J. Appl. Polym. Sci. 2019, 136, 47187. Jha, A.; Dutta, B.; Bhowmick, A. Effect of fillers and plasticizers on the performance of novel heat and oil-resistant thermoplastic elastomers from nylon-6 and acrylate rubber blends. J. Appl. Polym. Sci. 1999, 74, 1490-1501. Jha, A.; Bhowmick, A.K. Mechanical and dynamic mechanical thermal properties of heat- and oil-resistant thermoplastic elastomeric blends of poly(butylene terephthalate) and acrylate rubber. J. Appl. Polym. Sci. 2000, 78, 1001-1008. Hu, S.; Chen, S.; Zhao, X.; Guo, M.; Zhang, L. The shape-memory effect of hindered phenol (AO-80)/acrylic rubber (ACM) composites with tunable transition temperature. Materials 2018, 11, 2461. Qiu, Z.; Qin, C.; Qiu, J. Study on application of the blends of nitrile rubber with acrylate rubber in the coat-metal sealing gasket. Adv. Mater. Res. 2012, 393, 1438-1442. Wilfredo, Y.; Anna, S.; Nancy, Y.; Zbigniew, R. Design, synthesis, characterization and optimization of PTT-b-PEO copolymers: a new membrane material for CO2 separation. J. Membr. Sci. 2010, 362, 407-416. Buchard, A.; Bakewell, C.M.; Weiner, J.; Williams, C.K. Recent developments in catalytic activation of renewable resources for polymer synthesis. Organo Renewa. 2012, 39, 175-224. Willke, T. Biotechnological production of itaconic acid. Appl. Microbiol. Biotechnol. 2001, 56, 289-295. Lee, J.; Kim, H.; Choi, S.; Yi, J.; Lee, S. Microbial production of building block chemicals and polymers. Curr. Opin. Biotechnol. 2011, 22, 758-767. Choi, S.; Song, C.; Shin, J.; Lee, S. Biorefineries for the production of top building block chemicals and their derivatives. Metab. Eng. 2015, 28, 223-239. Wang, J.; Qian, W.; He, Y.; Xiong, Y.; Song, P.; Wang, R.M. Reutilization of discarded biomass for preparing functional polymer materials. Waste Manag. 2017, 65, 11-21. Dong, W.; Li, T.; Xiang, S.; Ma, P.; Chen, M. Influence of glutamic acid on the properties of poly(xylitol glutamate sebacate) bioelastomer. Polymers 2013, 5, 1339-1351. Dai, L.; Tai, C.; Shen, Y.; Guo, Y.; Tao, F. Biosynthesis of 1,4-butanediol from erythritol using whole-cell catalysis. Biocatal. Biotransform. 2019, 37, 92-96. Wang, G.; Jiang, M.; Zhang, Q.; Wang, R.; Liang, Q.; Zhou, G. New bio-based copolyesters derived from 1,4-butanediol, terephthalic acid and 2,5-thiophenedicarboxylic acid: synthesis, crystallization behavior, thermal and mechanical properties. Polym. Test. 2019, 75, 213-219. Wang, R.; Ma, J.; Zhou, X.; Wang, Z.; Kang, H.; Zhang, L.; Hua, K.C.; Kulig, J. Design and preparation of a novel cross-linkable, high molecular weight, and bio-based elastomer by emulsion polymerization. Macromolecules 2012, 45, 6830-6839. Wang, R.; Yao, H.; Lei, W.; Zhou, X.; Zhang, L.; Hua, K.C.; Kulig, J. Morphology, interfacial interaction, and properties of a novel bioelastomer reinforced by silica and carbon black. J. Appl. Polym. Sci. 2013, 129, 1546-1554. Wang, Z.; Zhang, X.; Zhang, L.; Tan, T.; Fong, H. Nonisocyanate biobased poly(ester urethanes) with tunable properties synthesized via an environment-friendly route. ACS Sustainable Chem. Eng. 2016, 4, 2762-2770. Okabe, M.; Lies, D.; Kanamasa, S.; Park, E.Y. Biotechnological production of itaconic acid and its biosynthesis in aspergillus terreus. Appl. Microbiol. Biotechnol. 2009, 84, 597-606. Kuenz, A.; Gallenmuller, Y.; Willke, T.; Vorlop, K.D. Microbial production of itaconic acid: developing a stable platform for high product concentrations. Appl. Microbiol. Biotechnol. 2012, 96, 1209-1216. Wei, T.; Lei, L.; Kang, H.; Qiao, B.; Wang, Z.; Zhang, L.; Coates, P.; Hua, K.C.; Kulig, J. Tough bio-based elastomer nanocomposites with high performance for engineering applications. Adv. Eng. Mater. 2012, 14, 1-2. Guo, B.; Chen, Y.; Lei, Y.; Zhang, L.; Zhou, W.; Rabie, A.B.; Zhao, J. Biobased poly(propylene sebacate) as shape memory polymer with tunable switching temperature for potential biomedical applications. Biomacromol. 2011, 12, 1312-1321. Tsujimoto, T.; Uyama, H. Full biobased polymeric material from plant oil and poly(lactic acid) with a shape memory property. ACS Sustainable Chem. Eng. 2014, 2, 2057-2062. Farmer, T.J.; Castle, R.L.; Clark, J.H.; Macquarrie, D.J. Synthesis of unsaturated polyester resins from various bio-derived platform molecules. Int. J. Mol. Sci. 2015, 16, 14912-14932. Gao, L.; Zheng, G.; Nie, X.; Wang, Y. Thermal performance, mechanical property and fire behavior of epoxy thermoset based on reactive phosphorus-containing epoxy monomer. J. Therm. Anal. Calorim. 2017, 127, 1419-1430. Wang, Z.; Wei, T.; Xue, X.; He, M.; Xue, J.; Song, M.; Wu, S.; Kang, H.; Zhang, L.; Jia, Q. Synthesis of fully bio-based polyamides with tunable properties by employing itaconic acid. Polymer 2014, 55, 4846-4856. Maisonneuve, L.; Lebarbe, T.; Grau, E.; Cramail, H. Structure-properties relationship of fatty acid-based thermoplastics as synthetic polymer mimics. Polym. Chem. 2013, 4, 5472. Lei, W.; Russell, T.P.; Hu, L.; Zhou, X.; Qiao, H.; Wang, W.; Wang, R.; Zhang, L. Pendant chain effect on the synthesis, characterization, and structure-property relations of poly(di-n-alkyl itaconate-coisoprene) biobased elastomers. ACS Sustainable Chem. Eng. 2017, 5, 5214-5223. Qiao, H.; Xu, W.; Chao, M.; Liu, J.; Lei, W.; Zhou, X.; Wang, R.; Zhang, L. Preparation and performance of silica/epoxy group-functionalized biobased elastomer nanocomposite. Ind. Eng. Chem. Res. 2017, 56, 881-889. Durant, Y.; Jiang, B.; Tsavalas, J. Emilsion polymerization of ester of itaconic acid. U. S. Pat. 2018, 932, 421. Chakraborty, S.; Ramakrishnan, S. Surface-Functionalized Polystyrene Latexes Using Itaconate-Based Surfmers. Langmuir 2018, 34, 11729−11737. Qiao, Z.; Qiu, T.; Liu, W.; Zhang, L.; Tu, J.; Guo, L.; Li, X. A “green” method for preparing ABCBA pentablock elastomers by using RAFT emulsion polymerization. Polym. Chem. 2017, 8, 3013. Schmidt, B.; Molinari, V.; Esposito, D.; Tauer, K.; Antonietti, M. Lignin-based polymeric surfactants for emulsion polymerization. Polymer 2017, 112, 418-426. Fernandez, A.M.; Held, U.; Willing, A.; Breuer, W.H. New green surfactants for emulsion polymerization. Prog. Org. Coat. 2005, 53, 246-255. Lei, W.; Qiao, H.; Zhou, X.; Wang, W.; Zhang, L.; Wang, R.; Hua, K.C. Synthesis and evaluation of bio-based elastomer based on diethyl itaconate for oil-resistance applications. Sci. China, Chem. 2016, 59, 1376-1383.

Round 2

Reviewer 2 Report

The revised version of the manuscript addresses all my concerns.

I only have two small things to be revised:

In Figure 4 CDCl3 should be CHCl3 In Figure 2, there should be a "co" between the bons of the different monomers. Otherwise it looks like a block copolymer

Author Response

The revised version of the manuscript addresses all my concerns.

I only have two small things to be revised:

In Figure 4 CDCl3 should be CHCl3

Answer: Thanks so much for the comment. We used deuterated chloroform (CDCl3) as the solvent. D stands for deuterated. We also provided some literatures for it.

[1] Gao, Y.; Li, Y.; Hu, X.; Wu, W.; Wang, Z.; Wang, R.; Zhang, L. Preparation and Properties of Novel Thermoplastic Vulcanizate Based on Bio-Based Polyester/Polylactic Acid, and Its Application in 3D Printing. Polymers 2017, 9, 694.

[2] Wang, R.; Ma, J.; Zhou, X.; Wang, Z.; Kang, H.; Zhang, L.; Hua, K.C.; Kulig, J. Design and preparation of a novel cross-linkable, high molecular weight, and bio-based elastomer by emulsion polymerization. Macromolecules 2012, 45, 6830-6839.

In Figure 2, there should be a "co" between the bons of the different monomers. Otherwise it looks like a block copolymer.

Answer: Thanks so much for the comment. We have corrected it in Figure 2.
